# The prevalence of insomnia and restless legs syndrome among Japanese outpatients with rheumatic disease: A cross-sectional study

**Kayoko Urashima[1], Kunihiro Ichinose[2]\*, Hideaki Kondo[3,4], Takahiro Maeda[3], Atsushi Kawakami[2], Hiroki Ozawa[1]**

1 Department of Neuropsychiatry, Medical and Dental Sciences, Nagasaki University Graduate School of Biomedical Sciences, Nagasaki, Japan, 2 Division of Advanced Preventive Medical Sciences, Department of Immunology and Rheumatology, Nagasaki University Graduate School of Biomedical Sciences, Nagasaki, Japan, 3 Department of General Medicine, Nagasaki University Graduate School of Biomedical Sciences, Nagasaki, Japan, 4 International Institute for Integrative Sleep Medicine, University of Tsukuba, Tsukuba, Japan

☯ These authors contributed equally to this work.
\* kichinos@nagasaki-u.ac.jp

**Data Availability Statement:** All relevant data are within the manuscript and its Supporting Information files.

## Abstract

The prevalence of symptomatic insomnia and the prevalence of restless legs syndrome (RLS) are known to be higher among patients with rheumatic diseases compared to the general population. The prevalences of insomnia and RLS reported in a questionnaire by Japanese patients with rheumatic diseases at an outpatient clinic were analyzed herein. The association between the patients' disease activity and their sleep quality was analyzed. Of 121 rheumatic disease patients, 70 were enrolled. The median (interquartile range) age at enrollment was 62.0 (47.8–68.0) years. There were 58 women (82.9%) and 12 men (17.1%), and 43 patients (61.4%) with rheumatoid arthritis (RA), nine (12.9%) with systemic lupus erythematosus (SLE), and 18 (25.7%) with other rheumatic diseases. Twenty patients (28.6%) had one or more moderate-to-severe insomnia symptoms, and 10 (14.3%) were diagnosed with RLS. Among the patients with RA, the swollen joint count based on a 28-joint assessment (SJC28) was significantly higher in the insomnia group (n = 13) compared to the non-insomnia group (n = 30) (p = 0.006). A classification and regression tree (CART) analysis showed that the cut-off points of ≥3 mg/day prednisolone (PSL) treatment and <16.54% as the transferrin saturation (TSAT) value would best predict RLS in rheumatic disease. Patients with rheumatic disease had a high prevalence of symptomatic insomnia and RLS. A higher dose of PSL and lower TSAT were associated with the occurrence of RLS.

## Introduction

The rheumatic diseases are a diverse group of chronic diseases characterized by the presence of chronic inflammation affecting structures of the musculoskeletal system, blood vessels, and other tissues. Sleep has a strong regulatory influence on immune functions. The disturbance of

**Funding:** This work was supported by a grant from the Nakatani Foundation for Advancement of Measuring Technologies in Biomedical Engineering (to K.I.).

**Competing interests:** The authors declare that they have no competing interests.

the body's immunity caused by sleep deprivation is not limited to the suppression of the response to pathogens; it can also result in the collapse of immunological self-tolerance, which can cause the onset of autoimmune disease [1–3]. Cross-sectional studies have shown that sleep disorders in individuals with autoimmune diseases are correlated with pain, fatigue, psychiatric manifestations, and disease activity [4].

The prevalence of symptomatic insomnia among rheumatic disease patients is as high as 47%–65% [5, 6], and the rates of patients' subjective complaints of deteriorated sleep quality are as high [5, 7–9]. Symptomatic insomnia is frequently associated with disease activity, with general symptoms such as arthralgia, and with various medications. However, insomnia sometimes persists even after an individual's original disease has improved [10]. The presence of insomnia increases the risk of developing rheumatic disease [1], and fragmentation of sleep is associated with exacerbation of the original disease [4, 11, 12].

Restless legs syndrome (RLS) may be present in 20%–30% of individuals' with rheumatic diseases [13–16]. RLS is accompanied by abnormal perception(s) such as an unusual, relaxing, or burning feeling or an impulsive feeling based mainly in the lower limbs. These symptoms become prominent at rest and are relieved by moving. Because RLS is often worse at night, RLS may induce severe insomnia, manifesting as difficulty initiating and maintaining sleep [17]. Although the prevalence of RLS in general populations in East Asian countries including Japan is known to be lower than those in Europe and the U.S. [18], the reported prevalence of RLS among Pakistani patients with rheumatic diseases was rather high at 19.1%.

We conducted the present study to determine the association between the disease activity of rheumatic diseases such as rheumatoid arthritis (RA) during patients' visits to an outpatient clinic and their sleep-related problems. The prevalence of RLS among Japanese outpatients with rheumatic diseases was also investigated.

## Patients and methods

### Patients

We collected 121 consecutive patients with rheumatic disease treated between August 2017 and May 2018 at the outpatient clinic at Nagasaki University Hospital. The quality of the studies was assessed on the basis of elements from the STROBE checklist for cross-sectional studies [19]. The eligibility criteria for patients participating in this study were: (1) having a clinical diagnosis of RA, systemic lupus erythematosus (SLE), mixed connective tissue disease (MCTD), systemic sclerosis, vasculitis, osteoarthritis, dermatomyositis, or spondyloarthritis by a rheumatologist; (2) a current user of at least prednisolone or an immunosuppressant according to their medical records; (3) >20 years old, and 4) able to provide informed consent. The exclusion criteria were: (1) the presence of any comorbidity of a medical or psychological/psychiatric condition or treatment revealed by a review of the patient's previous or outside medical records, in the opinion of the Principal Investigator, and (2) being unwilling to participate in a research study or provide research samples or data.

Seventy patients met the initial study criteria and were enrolled in the study (S1 Fig). The STROBE reporting guidelines were used in the design and implementation of our research [19]. The patients' subjective sleep quality, sleep disturbance, severity of insomnia, and daytime sleepiness were evaluated. The patients underwent a screening test for RLS by the questionnaire method described below. Patients who declined to provide informed consent to participate in the study were excluded. Written informed consent was obtained from all individual participants included in the study. The study was reviewed and approved by the Medical Ethical Committee of Nagasaki University Hospital (approval no. 16092619).

## Data collection

The patients' demographic and clinical characteristics were collected at the time of the RLS screening (Table 1). The demographic data included each patient's gender, age, body mass index (BMI), drinking habit, smoking habit, comorbidities, past history, and history of regular medication. The patients' laboratory data were obtained, including the white and red blood cell counts, hemoglobin, platelet counts, total protein, albumin, sodium, potassium, chlorine, aspartate aminotransferase (AST), alanine aminotransferase (ALT), lactate dehydrogenase, blood urea nitrogen, serum creatinine (Cr), estimated glomerular filtration rate (eGFR), serum iron (Fe), unsaturated iron binding capacity, transferrin saturation (TSAT), C-reactive protein (CRP), serum ferritin, blood sedimentation (represented by the erythrocyte sedimentation rate [ESR]), glucose, HbA1c, and urine protein/creatinine ratio (Up/Ucr).

For the RA patients, the following information was also obtained: the tender joint count based on a 28-joint assessment (i.e., the TJC28), the swollen joint count based on a 28-joint assessment (the SJC28), the Medical Doctors' Global Assessment (MDGA) of RA disease

**Table 1. The patients' demographic and clinical characteristics.**

| | |
|---|---|
| Age, yrs, median (IQR) | 62 (47.8–68) |
| Female, n (%) | 58 (82.9) |
| Diagnosis, n (%): | |
| RA | 43 (61.4) |
| SLE | 9 (12.9) |
| Mixed connective tissue disease | 5 (7.1) |
| Others | 13 (18.6) |
| Disease duration, yrs, median (IQR) | 11 (6–18) |
| BMI, kg/m$^2$, median (IQR) | 21.6 (19.9–24.4) |
| Smoker, n (%) | 10 (14.3) |
| Sleeping pills, n (%) | 10 (14.3) |
| Oral iron preparation, n (%) [*] | 1 (1.5) |
| Vitamin D preparations, n (%) [*] | 14 (20.6) |
| Antipsychotic drugs, n (%) [*] | 1 (1.5) |
| Laboratory data, median (IQR): | |
| Hemoglobin, g/dl | 12.8 (11.6–14.1) |
| BUN, mg/dl | 14.0 (11.0–17.0) |
| Creatinine, mg/dl | 0.69 (0.60–0.80) |
| eGFR, ml/min/1.73m$^2$ | 74.3 (58.4–85.2) |
| Total protein, g/dl | 7.50 (7.18–8.82) |
| Albumin, g/dl | 4.15 (3.90–4.43) |
| Ferritin, ng/ml | 55.0 (20.0–99.0) |
| Iron, μg/dl | 77.5 (54.0–118) |
| UIBC, μg/dl | 238 (200–303) |
| TSAT, % | 25.3 (14.6–38.2) |
| ESR, mm/hr | 12.0 (8.00–24.0) |
| CRP, mg/dl | 0.08 (0.03–0.20) |
| Up/Ucr, g/g▪Cr | 0.08 (0.05–0.15) |

[*] A total of 68 patients were evaluated. BMI: body mass index, BUN: blood urea nitrogen, CRP: C-reactive protein, eGFR: estimated glomerular filtration rate, ESR: erythrocyte sedimentation rate, IQR: interquartile range, RA: rheumatoid arthritis, SLE: systemic lupus erythematosus, TSAT: transferrin saturation, UIBC: Unsaturated iron binding capacity, Up/Ucr: urine protein/creatinine ratio.

activity on a 100-mm visual analog scale, and the Patient's Global Assessment (PGA) of activity on a 100-mm analog scale. Four clinical activity disease indices, i.e., the Disease Activity Score in 28 joints (DAS28)-CRP, the DAS28-ESR, the Simplified Disease Activity Index (SDAI), and the Clinical Disease Activity Index (CDAI) were calculated as described [20–22].

## Questionnaires

We evaluated the patients' sleep quality and disturbance retrospectively over a 1-month period by using the self-reported Japanese version of the Pittsburgh Sleep Quality Index (PSQI) [23]. The 19 questions are categorized into seven components, graded with scores that range from 0 to 3. The PSQI components are as follows: subjective sleep quality (C1), sleep latency (C2), sleep duration (C3), habitual sleep efficiency (C4), sleep disturbances (C5), use of sleeping medication (C6), and daytime dysfunction (C7). The PSQI global score (GS) is evaluated as 0 to 21 points, and a higher score indicates low sleep quality and sleep disturbance. The PSQI cut-off value for poor sleep quality was estimated at ≥6 points.

The severity of the patients' insomnia was assessed by the Japanese version of the Insomnia Severity Index (ISI) [24, 25]. The daytime sleepiness was assessed by the Japanese version of the Epworth Sleepiness Scale (ESS) [26, 27]. The total ISI scores are divided into seven items, each providing a five-point Likert scale (0 = no problem, 4 = very severe problem), and the items assess the severity of difficulty initiating sleep (DIS), difficulty maintaining sleep (DMS), and waking up too early (WE). A cut-off score of 10 has been used as the threshold for pathological conditions of insomnia [28]. Responders who had 'mild' to 'very severe' symptoms were defined as having DIS, DMS, and/or WE. Responders who had 'moderate' to 'very severe' symptoms were defined as having moderate to severe DIS, DMS, and/or WE. The insomnia group was defined herein as having at least moderate to severe DIS, DMS, and/or WE. The total ESS score ranges from 0 to 24 points, with higher scores indicating stronger subjective daytime sleepiness. ESS scores of ≥10 points represent increasing levels of 'excessive daytime sleepiness.'

For the screening test for RLS, the Japanese version of the Cambridge-Hopkins questionnaire short-form 13 (CH-RLSq13) was used [29, 30]. The CH-RLSq13 is a self-administered questionnaire with high sensitivity and specificity for the diagnosis of RLS. The questionnaire is comprised of 13 items, 10 of which are related to characteristic symptoms and the exclusion of other conditions (e.g., leg cramping, positional discomfort). The remaining three items are related to the severity and onset of symptoms.

Based on the patients' data obtained from the sleep-related questionnaire, the patients were divided into the poor-sleeper group (with a PSQI global score of ≥6 points) and the nonpoor-sleeper group. The patients were also divided into the severe-insomnia group (with an ISI total score of ≥10 points) and the nonsevere-insomnia group. A third division of patients was made: the excessive daytime sleepiness (EDS) group (with an ESS total score of ≥10 points) and the non-EDS group.

## Diagnosis of RLS

After the questionnaire screening was conducted, a board-certified physician of the Japanese Society of Sleep Research (H.K.) interviewed the patients who were suspected of having RLS. These patients were screened by a questionnaire for the five essential diagnostic criteria for RLS as defined by the International Restless Legs Syndrome Study Group (IRLSSG) [17].

## Multiplex cytokine/chemokine bead assays

We performed a multiplex analysis of undiluted serum supernatants using the Milliplex MAP Human Cytokine/Chemokine Panel 1 Pre-mixed 38 Plex (Merck Millipore, Darmstadt,

Germany) according to the manufacturer's instructions. Within 30 min of their collection, the serum samples were centrifuged at 1500 rpm at 4˚C for 5 min, and the liquid phase of the serum was stored at −80˚C until use. The levels of the following 41 cytokines/chemokines in the liquid phase of the serum were measured: vascular endothelial growth factor (VEGF), tumor necrosis factor-beta (TNF-β), tumor necrosis factor-alpha (TNF-α), transforming growth factor (TGF)-α, macrophage inflammatory protein (MIP)-1β, MIP-1α, myeloid dendritic cells (MDCs)/ C-C Motif Chemokine Ligand 22 (CCL22), monocyte chemotactic protein (MCP)-3, MCP-1, interferon-gamma (IFN-γ)-induced protein (IP)-10, interleukin (IL)-17, -15, -13, -12 (p70), -12 (p40), -10, -9, -8, -7, -6, -5, -4, -3, -2, -1ra, -1β, and -1α, IFN-γ, IFN-α2, growth-related cytokine (GRO), granulocyte macrophage-colony stimulating factor (GM-CSF), granulocyte-colony stimulating factor (G-CSF), fractalkine, Flt-3 ligand, fibroblast growth factor (FGF)-2, eotaxin, epidermal growth factor (EGF), and sCD40L.

The concentrations were calculated based on the respective standard curve for each cytokine's/chemokine's concentration, assayed in the same manner as the serum samples. The detection limit for each molecule was determined by the recovery of the corresponding standard, and the lowest values with >70% recovery were set as the lower detection limits. All samples were analyzed in duplicate.

## Statistical analyses

Categorical variables are presented as counts and percentages. Continuous variables are presented as medians and interquartile ranges (IQR) where non-normally distributed. A nonparametric Wilcoxon rank sum test was used for the inter-group comparisons of multiple variables. Fisher's exact test was also used to test the possible association between each variable factor and the subgroups. Decision tree models for predicting RLS were built with the Classification and Regression Trees (CART) algorithm [31]. The statistical analyses were performed using JMP® Pro14 software (SAS Institute, Cary, NC).

## Results

### The prevalence and insomnia symptoms in rheumatic disease

The prevalence and severity of insomnia in rheumatic disease were determined: the patients' median score on the PSQI was 6 (IQR 4–9), showing multimodality without normal distribution, and higher scores indicate that the patient's sleep quality is severely disturbed (Fig 1). There were 36 patients (51.4%) who had poor sleep. The number of patients who had DIS, DMS, and WA were 30 (42.9%), 28 (40%), and 20 (28.6%), respectively (Table 2). The number of patients with any insomnia symptoms was 38 (54.3%). Twenty patients (28.6%) had one or more moderate-to-severe insomnia symptoms.

### Comparison of the RA patients' disease activity scores between insomnia group and non- insomnia group

In the group of 43 RA patients, the disease activity scores were compared between the patients with and without any moderate-to-severe insomnia symptoms: the SJC28 was significantly higher in the insomnia group (n = 13) compared to the non-insomnia group (n = 30) (p = 0.006) (S1 Table).

### The prevalence of RLS and its disease-related features in rheumatic disease

Ten patients (14.3%) with rheumatic disease were diagnosed with RLS, including five RA patients, two SLE patients, and one patient with MCTD. The median age (minimum–

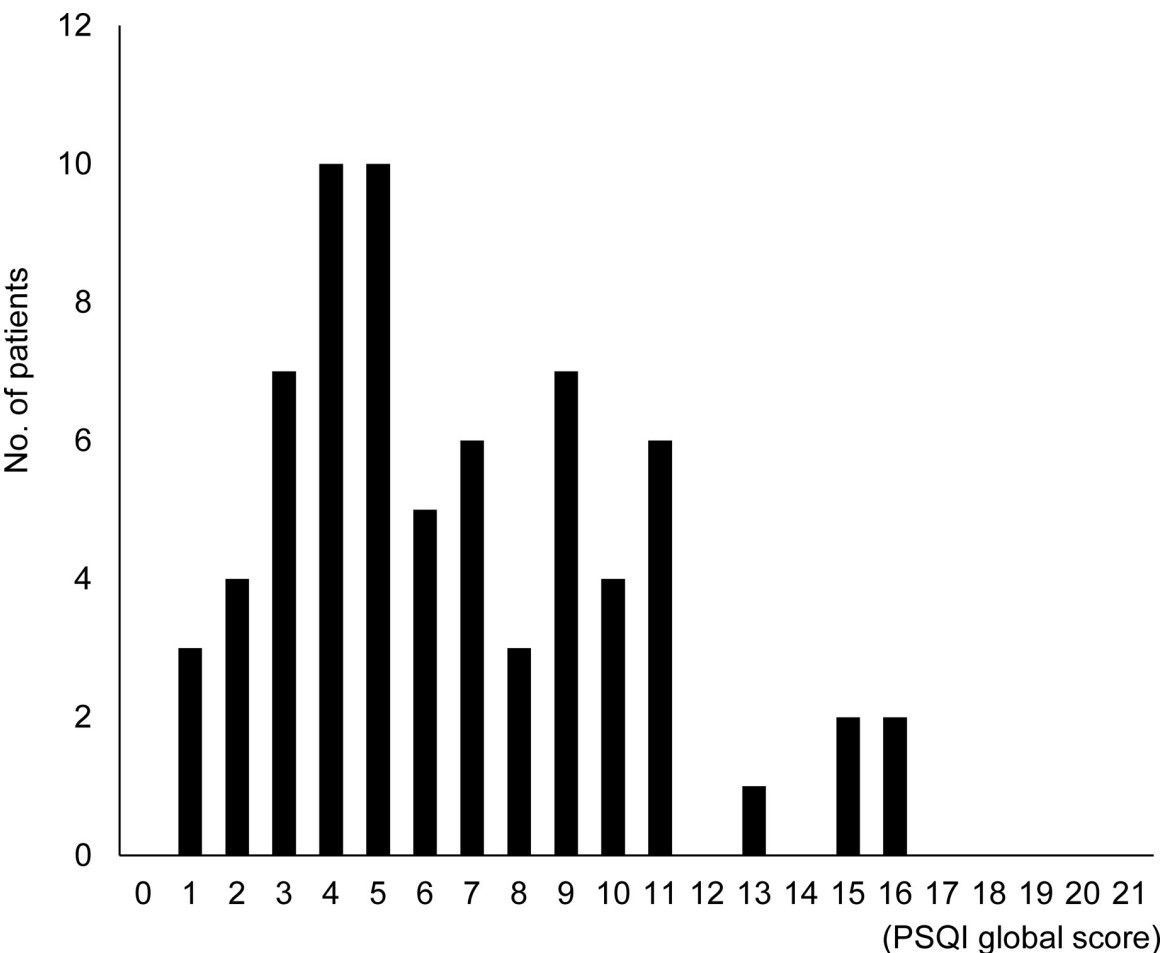

**Fig 1. Histogram of the patients' Pittsburgh Sleep Quality Index global scores.**

maximum) at RLS onset was 40 (15–60) years. Five of the 10 patients with RLS developed RLS after the diagnosis of rheumatic disease: in three of these five patients, the RLS and rheumatic disease had developed at almost the same time; in one patient, the RLS had developed before the diagnosis of rheumatic disease; in the remaining patient, the onset of RLS was not known. Nine patients reported experiencing RLS symptoms <2 days/week during the prior 12 months, and only one patient reported experiencing RLS symptoms 4–5 days/week.

We divided the total series of 70 patients into two groups based on whether they had RLS or not (Table 3). Among the disease-related features at baseline, younger age (p = 0.02), higher PSL dose (p = 0.002), lower total protein (p = 0.02), lower serum level of iron (p = 0.03) and lower TSAT (p = 0.03) were significantly related to RLS.

## Comparison of the patients' backgrounds evaluated with the PSQI, ISI, and ESS in the RLS and non-RLS groups

The patients' backgrounds evaluated with the PSQI, ISI, and ESS were compared between the RLS group and the non-RLS group (Table 4). Eight of the 10 patients (80%) in the RLS group and 19 of the 60 patients (32%) in the non-RLS group reported getting <6 hr of sleep per night, revealing a significant between-group difference (p = 0.004). There was no significant difference regarding difficulty sleeping in seven of the RLS patients (70%) compared to 23 of

**Table 2. The patients' sleep characteristics.**

| | |
|---|---|
| PSQI GS, median (IQR) | 6 (4–9) |
| PSQI GS ≥6 | 36 (51.4) |
| ISI, median (IQR) | 4 (2–6) |
| ISI ≥10, n (%) | 10 (14.3) |
| Symptoms of insomnia, n (%) | |
| DIS, mild to very severe | 30 (42.9) |
| DIS, moderate to very severe | 13 (18.6) |
| DMS, mild to very severe | 28 (40.0) |
| DMS, moderate to very severe | 12 (17.1) |
| WA, mild to very severe | 20 (28.6) |
| WA, moderate to very severe | 1 (1.4) |
| Any symptoms | 38 (54.3) |
| Any moderate to severe symptoms | 20 (28.6) |
| ESS, median (IQR) | 5 (3–8) |
| ESS ≥10, n (%) | 13 (18.6) |
| RLS, n (%) | 10 (14.3) |

DIS: difficulty initiating sleep, DMS: difficulty maintaining sleep, ESS: Epworth Sleepiness Scale, ISI: Insomnia Severity Index, PSQI GS: Pittsburgh Sleep Quality Index global score, RLS: restless legs syndrome, WA: wake up too early.

the non-RLS patients (38%) (p = 0.06). There was no significant between-group difference in daytime sleepiness assessed by the ESS.

## Assessment of the cut-off values of the predictors of RLS in the rheumatic disease patients

The predicting factors associated with RLS in the rheumatic disease patients as revealed by the CART analysis were examined, with the explanatory variables of age at onset, the daily PSL dose, and the TP, Fe, and TSAT levels (which showed significant differences in a univariate analysis) for the CART analysis. In the decision tree analysis, the calculated minimum number of cases of the child node after the analysis was 10, and the analysis was terminated before the number of cases of the child node became <10. The cut-off point of ≥3 mg/day for the PSL dose and <16.54% for TSAT provided the best performance for predicting RLS in rheumatic disease (Fig 2).

## Discussion

This study provides the first report of the prevalence of RLS among individuals with rheumatic diseases in Japan. The association between RLS and higher PSL dose in rheumatic diseases has not been discussed, and it is necessary to clarify the significance of this association in pathophysiology; a verification of our findings in further prospective studies of larger multicenter populations is also needed.

The prevalence of RLS among our present patients with rheumatic diseases was significantly higher than that reported in the general populations of Asian countries [18]. The reported prevalence of RLS in the Japanese general population is 1.8% [32]. The reported prevalence of RLS was 27.7% among RA patients in Canada [14], 15.3% among patients with Sjögren's syndrome in Sweden [33], 30.6%–37.5% among subjects with SLE in Turkey and Canada [15, 34], and 19.1% among patients with rheumatic diseases in Pakistan [16].

**Table 3. Comparisons of demographic and clinical characteristics between RLS group and non-RLS group.**

| | RLS (n = 10) | Non-RLS (n = 60) | p-value |
|---|---|---|---|
| Age, yrs, median (IQR) | 47.5 (36.0–64.0) | 62.5 (49.3–71.0) | 0.02 |
| Female, n (%) | 8 (80.0) | 50 (83.3) | 0.80 |
| Diagnosis, n (%) | | | |
| RA | 5 (50.0) | 38 (63.3) | 0.42 |
| SLE | 2 (20.0) | 7 (11.7) | 0.61 |
| Disease duration, yrs, median (IQR) | 15 (7.5–23) | 10 (5–18) | 0.25 |
| BMI, kg/m$^2$, median (IQR) | 21.2 (19.0–27.9) | 21.9 (19.9–24.3) | 0.87 |
| Smoker, n (%) | 9 (90.0) | 51 (85.0) | 0.68 |
| Sleeping pills, n (%) | 2 (20) | 8 (13.3) | 0.63 |
| Vitamin D preparations, n (%) * | 3 (30) | 11 (19.0) | 0.42 |
| Prednisolone (mg) | 6.00 (3.75–8.25) | 0.50 (0.00–0.50) | 0.002 |
| Prednisolone ≥ 3 mg (%) | 9 (90.0) | 20 (33.3) | 0.001 |
| Laboratory data, median (IQR): | | | |
| Hemoglobin, g/dl | 12.8 (11.7–13.9) | 12.7 (11.6–14.15) | 0.78 |
| BUN, mg/dl | 12.0 (7.0–20.0) | 14.0 (12.0–17.0) | 0.28 |
| Creatinine, mg/dl | 0.63 (0.53–0.83) | 0.7 (0.6–0.8) | 0.30 |
| eGFR, ml/min/1.73m$^2$ | 82.0 (74.3–97.5) | 72.4 (58.0–95.3) | 0.06 |
| Total protein, g/dl | 7.2 (7.1–7.4) | 7.6 (7.2–7.9) | 0.02 |
| Albumin, g/dl | 4.05 (3.8–4.325) | 4.2 (4.0–4.5) | 0.17 |
| Ferritin, ng/ml | 23.5 (14.0–64.5) | 55.0 (24.5–104) | 0.06 |
| Iron, µg/dl | 54.0 (35.0–82.0) | 84.5 (56.3–128) | 0.03 |
| Unsaturated iron binding capacity, µg/dl | 292 (261–329) | 226 (197–294) | 0.10 |
| TSAT, % | 14.7 (10.6–25.8) | 26.5 (17.0–39.0) | 0.03 |
| ESR, mm/hr | 10.0 (7.5–27.3) | 12.0 (8.0–24.0) | 0.92 |
| CRP, mg/dl | 0.16 (0.03–0.60) | 0.08 (0.03–0.19) | 0.48 |
| Up/Ucr, g/g・Cr | 0.04 (0.06–0.13) | 0.08 (0.05–0.17) | 0.25 |

BMI: body mass index, BUN: blood urea nitrogen, CRP: C-reactive protein, eGFR: estimated glomerular filtration rate, ESR: erythrocyte sedimentation rate, IQR: interquartile range, RA: rheumatoid arthritis, SLE: systemic lupus erythematosus, TSAT: transferrin saturation, Up/Ucr: urine protein/creatinine ratio. P-values were determined by nonparametric Wilcoxon rank sum test and Fisher's exact test.

Compared to previous reports, the prevalence of RLS in our study (14.3%) was similar to that in other countries.

Our study subjects were outpatients with relatively stable conditions. There have been no reports of RLS coexistence in the early stages of or in conditions with high disease activity. Interestingly, half of our patients with RLS developed RLS after the diagnosis of rheumatic disease; in three of these five patients, the RLS and rheumatic disease had developed almost simultaneously; in one patient, the RLS developed before the diagnosis of rheumatic disease, and in the remaining patient, the onset of RLS was unknown. It is necessary to determine whether RLS develops secondarily to rheumatic disease, and to further clarify the association between the disease activity of rheumatic disease and RLS.

It is not yet known whether PSL treatment affects the disease development of RLS in rheumatic disease. Only a single case report showed that oral PSL was effective in a patient with RLS who was resistant to other therapies [35]. However, other reports described an association between PSL treatment and sleep disorder [2, 36]. In our present investigation, the serum levels of GRO (p = 0.01), IL-8 (p = 0.003), and IL-1ra (p = 0.03) measured by multiplex cytokine/

**Table 4. Comparisons of sleep characteristics between the RLS and non-RLS groups.**

|  | RLS (n = 10) | Non-RLS (n = 60) | p-value |
|---|---|---|---|
| PSQI |  |  |  |
| Global score, median (IQR) | 8.0 (4.0–14.0) | 5.5 (4.0–9.0) | 0.16 |
| Global score ≥6, n (%) | 6 (60) | 30 (50) | 0.56 |
| PSQI components, n (%) |  |  |  |
| C1: subjective sleep quality ≥2 | 3 (30) | 14 (23) | 0.64 |
| C2: sleep latency ≥2* | 4 (40) | 15 (25) | 0.32 |
| C3: sleep duration ≥2** | 8 (80) | 19 (32) | 0.004 |
| C4: habitual sleep efficiency ≥2*** | 1 (10) | 4 (6.7) | 0.71 |
| C5: sleep disturbances ≥2 | 7 (70) | 37 (62) | 0.61 |
| C6: use of sleeping medication ≥2**** | 2 (20) | 12 (20) | 1.00 |
| C7: daytime dysfunction ≥2 | 1 (10) | 3 (5.0) | 0.53 |
| ISS, median (IQR) | 5.5(3.0–8.5) | 4.0 (2.0–6.0) | 0.25 |
| ISS ≥10, n (%) | 2 (20.0) | 8 (13.3) | 0.63 |
| Symptoms of insomnia, n (%) |  |  |  |
| Difficulty initiating sleep | 7 (70) | 23 (38) | 0.06 |
| Difficulty maintaining sleep | 4 (40) | 24 (40) | 1.00 |
| Waking up too early | 1 (10) | 19 (31) | 0.16 |
| Any symptoms, mild to very severe | 7 (70) | 31 (52) | 0.28 |
| ESS, median (IQR) | 5.0 (3.25–7.75) | 5.0 (3.00–8.75) | 0.93 |
| ESS ≥10, n (%) | 2 (20.0) | 11 (18.3) | 1.00 |

*Sleep latency ≥31 min and the presence of difficulty initiating sleep.

**Total sleep time <6 hr.

***Sleep efficiency <85%.

**** >1 time. P-values were determined by nonparametric Wilcoxon rank sum test and Fisher's exact test. ESS: Epworth Sleepiness Scale, ISS: Insomnia Severity Index, PSQI: Pittsburgh Sleep Quality Index.

chemokine bead assays (S2 Table) were significantly higher in the group with the PSL cut-off point of ≥3 mg/day compared to the PSL <3 mg/day group, suggesting potential roles of inflammation and immunological alteration in the mechanisms of action in RLS [37].

The distances between each pair of cytokines were also analyzed, based on the Spearman's correlation coefficients of the PSL cut-off point ≥3 mg/day; the results are shown as a heatmap (S2 Fig). The correlations of pairs of cytokines/chemokines/growth factors showed different patterns between the PSL ≥3 mg/day group and the PSL <3 mg/day group. The cytokines/chemokines/growth factors including pro-inflammatory cytokines (IL-2, IL-6, IL-15, IFN-γ) were generally lower and correlated with an anti-inflammatory cytokine (IL-1ra) with PSL <3 mg/day group compared to the PSL ≥3 mg/day group. This result may indicate that an anti-inflammatory cytokine might exert control in the PSL <3 mg/day group as a host defense mechanism. Accordingly, our PSL ≥3 mg/day and <3 mg/day groups differed with elevated downstream levels of pro-inflammatory cytokines/chemokines/growth factors, and these variations may have affected the disease activity and mechanism of RLS.

Iron deficiency is associated with RLS pathogenesis, and iron replacement therapy is generally widely used as a treatment for RLS when the patient's level of TSAT, a marker of iron availability, is low [38]. In the present study, the TSAT level was significantly lower in the RLS group (p = 0.03). These data suggest that when iron deficiency is present in a patient with a rheumatic disease, iron replacement therapy is effective. However, under chronic inflammation, increased iron trapping within the macrophages and hepatocytes results in a higher level

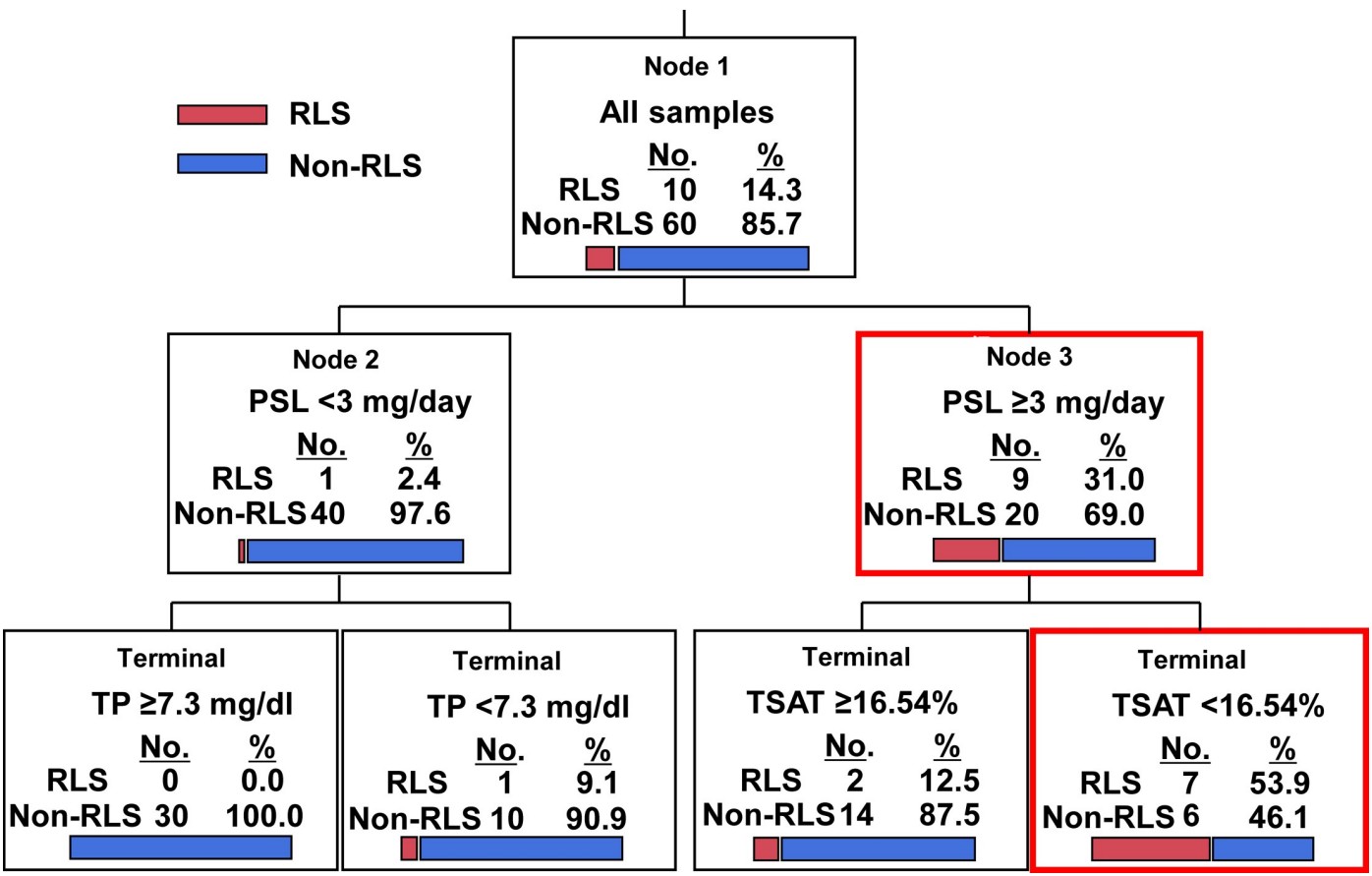

**Fig 2. Classification and regression trees (CART) for predicting factors associated with RLS.**

of ferritin [39, 40], and clinicians should be aware that the administered iron may not be effective for erythropoiesis.

The age of the present RLS group was significantly lower than that of the non-RLS group (p = 0.02). In this study, five of the 10 patients in the RLS group developed RLS before the age of 45 years. It was reported that early-onset RLS (occurring before the age of 45 years) has a high incidence of family history [41, 42]. In the present examination, one or more genetic factors may be involved with the earlier onset of RLS; however, our patients' family histories of RLS were not investigated sufficiently.

RLS often results in a higher prevalence of DIS and DMS compared to subjects without RLS [17]. In our study, the quality of sleep tended to be worse in the RLS group, but it was not clearly related to insomnia. This may be due to the fact that seven of the 10 RLS patients did not experience the symptoms of insomnia every day. In addition, only one RLS patient felt moderately distressed, although the patient was aware of RLS symptoms ≥4 days/week. We speculate that insomnia and RLS had a poor correlation in our study population in part because the severity of RLS was mild.

Vitamin D deficiency has been reported to be associated with RLS, and vitamin D replacement therapy has been effective in reducing the symptoms of RLS patients who had vitamin D deficiency [43–46]. In the present study, 20.6% of the enrolled patients were administered vitamin D, and this therapy may have obscured symptoms of RLS. It is necessary to make an adjustment that includes vitamin D deficiency and replacement therapy.

It was reported that insomnia could be attributed to RA in up to 42% of cases, linking sleep disturbance to pain, mood, and disease activity [47]. Our present findings also showed that the disease activity component of RA was higher in the insomnia group compared to the non-insomnia group. The results of our analyses indicated that insomnia is a consequence of joint pain in patients with RA, and chronic joint pain may result in a vicious cycle in which sleep disturbance activates clinical symptoms of pain, which then contribute to further sleep loss.

The limitations of our study deserve some discussion. First, the study population consisted of a small number of outpatients (n = 70) treated at a single center, with no a-priori statistical power analysis. This was an exploratory study, and we did not determine the number of samples required at the start of the study because the prevalence of RLS in Japanese rheumatic diseases is unknown. Second, there was a potential source of bias at the baseline including confounding (per the STROBE checklist). Because the number of RLS patients in this study was small and a multivariate analysis could not be performed, the adjustment for confounding factors was insufficient. In addition, patients with various rheumatic diseases such as RA and SLE were included in this study, and thus the patients' background characteristics such as being at a susceptible age for a disease and the amount of PSL and concomitant immunosuppressants varied depending on the rheumatic disease. The study population was thus not homogeneous with respect to the patients' backgrounds and treatments, which are likely to affect insomnia and RLS. Additional similar investigations based on each rheumatic disease would be informative.

Third, patients with high disease activity could not be included because this study was of outpatients with stable symptoms, and thus patients with relatively mild symptoms of insomnia and RLS might have participated. However, all of the data are presented in order to avoid selective reporting. In this study, an analysis was performed based on a hypothesis, and there is no evidence of data dredging. Further prospective studies of larger multicenter populations are required to clarify and sufficiently adjust the analyses for RLS and related factors in rheumatoid diseases, including confounding factors.

## Conclusions

This is the first report of the prevalence of insomnia and RLS in Japanese patients with rheumatic diseases. The prevalence of RLS in this series of patients with rheumatic diseases was 14.3%, and we consider this value as simply exploratory due to the limitations of the study. The cut-off point of $\geq$3 mg/day for the PSL dose and <16.54% for TSAT were observed to provide the best performance for predicting RLS in rheumatic disease. The serum levels of GRO, IL-8, and IL-1ra were significantly higher in the group with the PSL cut-off point of $\geq$3 mg/day compared to the PSL <3 mg/day group, suggesting potential roles of inflammation and immunological alteration in the mechanisms of action in RLS. The relationships among rheumatic disease activity with an imbalance of cytokines/chemokines, RLS, and insomnia should be clarified in large multicenter populations.

## Supporting information

**S1 Fig. Patient enrollment flow chart.**
(TIF)

**S2 Fig. Heat-maps of nearest-neighbor correlations of cytokines in the PSL cut-off point $\geq$3 mg/day and <3 mg/day.** For each cytokine analyzed, the distance between the PSL $\geq$3 mg/day group and the PSL <3 mg/day group was determined based on Spearman's correlation coefficient.
(TIF)

**S1 Table. Insomnia and disease activity scores in the studied RA patients.**
(DOCX)

**S2 Table. Cytokine profile of patients treated with prednisolone (PSL) ≥3 mg/day and those treated with <3 mg/day.**
(DOCX)

**S1 Data.**
(XLSX)

## Author Contributions

**Conceptualization:** Kunihiro Ichinose, Hideaki Kondo.

**Data curation:** Kayoko Urashima, Kunihiro Ichinose.

**Formal analysis:** Kunihiro Ichinose.

**Funding acquisition:** Kunihiro Ichinose, Atsushi Kawakami, Hiroki Ozawa.

**Investigation:** Hideaki Kondo.

**Methodology:** Hideaki Kondo.

**Project administration:** Takahiro Maeda.

**Resources:** Kunihiro Ichinose, Hiroki Ozawa.

**Supervision:** Kunihiro Ichinose, Hideaki Kondo, Takahiro Maeda, Atsushi Kawakami, Hiroki Ozawa.

**Validation:** Takahiro Maeda, Atsushi Kawakami, Hiroki Ozawa.

**Visualization:** Atsushi Kawakami, Hiroki Ozawa.

**Writing – original draft:** Kayoko Urashima, Kunihiro Ichinose.

**Writing – review & editing:** Kunihiro Ichinose, Hideaki Kondo.

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
