## [Decision Letter · Decision Letter 0]

23 Dec 2019

PONE-D-19-29396

The prevalence of insomnia and restless legs syndrome among Japanese outpatients with rheumatic disease: A cross-sectional study

PLOS ONE

Dear Dr. Ichinose,

Thank you for submitting your manuscript to PLOS ONE. After careful consideration, we feel that it has merit but does not fully meet PLOS ONE’s publication criteria as it currently stands. Therefore, we invite you to submit a revised version of the manuscript that addresses the points raised during the review process.

In particular, your manuscript was assessed by four reviewers, all experts in sleep and RLS research. I encourage you to revise your manuscript taking account of all the comments made by each reviewer. Please, pay particular attention to highlight the limitations of your manuscript mentioned by reviewers 2 and 3. 

In addition, please take account of the following:

- paragraph on patients, page 5-6. Please, specify the "initial study criteria", and in particular the eligibility criteria and the sources and methods of selection of participants, according to the STROBE checklist. Were the patients consecutive?

- please, present differences with P >= 0.05 as non-significant, rather than as trends or tendencies. Please, provide a reference for the Classification and Regression Trees (CART) algorithm. Please, avoid making unsupported inferences about the significance of differences between your study sample and the general population (in this respect, cf. also the comment by reviewer 3 on the lack of a control group in your study).

- In the Discussion section, please describe your efforts to address potential sources of bias, as per the STROBE checklist, addressing in particular confounding variables, lack of generalizability, selective reporting, post hoc analyses, and data dredging. In the paragraph on limitations (page 18), please clarify or correct the sentence "We examined a small number of outpatients (n=41) treated at a single center".

- Please, cite and discuss the following reference: Mod Rheumatol. 2013 Jul;23(4):705-8. doi: 10.1007/s10165-012-0714-x. Epub 2012 Jul 14, which is related to RLS and rheumatic diseases in Asia.

- The last paragraph of the Discussion (page 19) needs be rewritten. The sentence "The patients ... were associated with RLS" is written incorrectly. The following sentence would have been valid even irrespective of  the present manuscript, and would thus seem to support the third reviewer's concern that this manuscript is not characterized by a sufficient level of novelty for the advancement of our knowledge in this field. Please, elaborate and clarify.

- Please, increase the resolution of supplementary figure 2.

- PLOS journals require authors to make all data necessary to replicate their study’s findings publicly available without restriction at the time of publication (cf. https://journals.plos.org/plosone/s/data-availability). You are therefore required to submit the values behind the median and interquartile ranges and other measures reported and the values used to build graphs, either in a public repository (recommended) or as supplementary files.

We would appreciate receiving your revised manuscript by Feb 04 2020 11:59PM. To enhance the reproducibility of your results, we recommend that if applicable you deposit your laboratory protocols in protocols.io, where a protocol can be assigned its own identifier (DOI) such that it can be cited independently in the future. For instructions see: http://journals.plos.org/plosone/s/submission-guidelines#loc-laboratory-protocols

We look forward to receiving your revised manuscript.

Kind regards,

Alessandro Silvani, M.D., Ph.D.

Academic Editor

PLOS ONE

Journal Requirements:

2. Please provide additional details regarding participant consent to collect demographic data, clinical data and serum samples. In the ethics statement in the Methods and online submission information, please ensure that you have specified (1) whether consent was informed and (2) what type you obtained (for instance, written or verbal, and if verbal, how it was documented and witnessed). If your study included minors, state whether you obtained consent from parents or guardians. If the need for consent was waived, please ensure that you have discussed whether all data were fully anonymized before you accessed them and/or whether the IRB or ethics committee waived the requirement for informed consent.

3. We noticed minor instances of text overlap with the following previous publication(s), which need to be addressed:

https://www.sciencedirect.com/science/article/abs/pii/S1521661618304297?via%3Dihub

The text that needs to be addressed involves the last paragraph of the Discussion section.

In your revision please ensure you cite all your sources (including your own works), and quote or rephrase any duplicated text outside the methods section. Further consideration is dependent on these concerns being addressed.

4. Please include additional information regarding the survey or questionnaire used in the study and ensure that you have provided sufficient details that others could replicate the analyses. For instance, if you developed a questionnaire as part of this study and it is not under a copyright more restrictive than CC-BY, please include a copy, in both the original language and English, as Supporting Information.

5. In your Methods section, please provide additional information about the participant recruitment method and the demographic details of your participants.

Please ensure you have provided sufficient details to replicate the analyses such as:

a) the recruitment date range (month and year),

b) a description of how participants were recruited, and

c) descriptions of where participants were recruited and where the research took place.

Reviewers' comments:

Reviewer's Responses to Questions

**Comments to the Author**

1. Is the manuscript technically sound, and do the data support the conclusions?

Reviewer #1: Yes

Reviewer #2: Partly

Reviewer #3: Yes

Reviewer #4: Yes

2. Has the statistical analysis been performed appropriately and rigorously? 

Reviewer #1: Yes

Reviewer #2: I Don't Know

Reviewer #3: Yes

Reviewer #4: Yes

3. Have the authors made all data underlying the findings in their manuscript fully available?

Reviewer #1: Yes

Reviewer #2: Yes

Reviewer #3: Yes

Reviewer #4: Yes

4. Is the manuscript presented in an intelligible fashion and written in standard English?

Reviewer #1: Yes

Reviewer #2: Yes

Reviewer #3: No

Reviewer #4: Yes

5. Review Comments to the Author

Reviewer #1: In the present study the authors intend to determine the association between patients'

disease activity of rheumatoid arthritis (RA) during their outpatient visits and their sleep-related problems. They investigated the prevalence of RLS among Japanese outpatients (121) with rheumatic diseases.

This was a cross-sectional survey without intervention of 70 patients with

rheumatic disease among 121 eligible outpatients.

Patients' subjective sleep quality, sleep disturbance, severity of insomnia, and daytime sleepiness has been evaluated and a screening test for RLS performed by questionnaire.

Comments:

Inclusion and exclusion criteria should be detailed.

STROBE guidelines must be explained or related to a citation.

The authors correctly approached the queries by an adequate analysis of both scores and laboratory data although the sample size is quite small to generalize the results.

The authors did a good examination of the literature and clearly displayed limits of the study.

the study is well written and contextualized, but not particularly original in the hypothesis and therefore in the conclusions.

in fact, these are clinic-lab data observations that confirm the general literature data applied to rheumatoid arthritis

Reviewer #2: This study by Kayoko Urashima et al, assessed the frequency of insomnia and restless legs syndrome (RLS) in consecutive outpatients with rheumatic diseases (RD). Although the small sample of patients enrolled, this study could be of interest, as insomnia and RLS are both prevalent in patients with RD.

Here are major concerns:

1- The definition of insomnia (primary endpoint of this study) is not clear. Did patients had face to face interview? Was ICSD-3 or DSM-5 criteria used for the diagnostic of insomnia?

2- In the method section, authors defines clinically severe insomnia with threshold of insomnia severity index (ISI) above 10/28, instead of >14/28, the validated cut-off for moderate insomnia, and the cutoff of 22/28 for severe insomnia. Can authors justify why they used different ISI cutoff for this study?

3- The international RLS study group rating scale (IRLSQ) is worldwide used to assess RLS severity. Did authors assess severity with IRLSQ? If not available, is there any data on RLS severity assessment (Clinical global impression)?

Here are minor concerns:

1- In the introduction, it’s stated that sleep deprivation can cause the onset of autoimmune diseases, however, the study cited is on animal model, can authors provide other clincial studies ?

2- The presence of RLS was well done, as RLS was screened with questionnaire, and the diagnostic confirmed by physician, in this population with chronic pain complain, a condition that can mimic RLS symptoms. Can authors provide how many patients met the five essential criteria, and how many have RLS Diagnostic?

3- What was the age at RLS onset and age at daily RLS symptoms? Some patients didn’t had daily RLS symptoms, can authors provide the frequency of RLS for each patients? Did authors considered patients with less than 3 time a week of RLS symptom as sufferer?

4- In case of ferritin below 75µg/l, RLS is considered as secondary to iron depletion. Can authors provide how many patients with RLS were below this cutoff ? is there any difference between RLS and no-RLS patients ?

4- Patients with comorbid RLS and RD were younger than those without RLS. However, RLS frequency increase with age in the general population. How authors explain such result?

5- In the discussion section, there is mistake : “Interestingly, five of the 10 patients in the present RLS group had developed RLS after the diagnosis of rheumatic disease: one patient had developed RLS during almost the same period, and three patients developed rheumatic disease before the onset of RLS (the time course of the fifth RLS patient was unknown).”

How many patients have RLS before RD diagnostic, and how many after ?

6- In the discussion section, this sentence is not clear: “The correlation of pairs of cytokines/chemokines/growth factors showed different patterns between the PSL ≥3 mg/day group and the PSL <3 mg/day group. The correlation of pro- and anti-inflammatory cytokines in serum varied between the PSL ≥3 mg/day and <3 mg/day groups, and these variations may have affected the disease activity and mechanism of RLS.”

Please, can authors explain with more details what is the clinical implication of such finding?

Reviewer #3: While the study has been carried out with sufficient accuracy and especially the doagnosis of RLS was made following appropriate procedures and criteria, this stdy suffers of some problems:

1) The sample size is quite small for a study of this nature:

2) The authors should have included a control group randomly selected from the general population of the same geographical region;

3) The English form needs improvement.

Finally, I doubt that describing an already well.know association (RLS and rheumatic disease) in a different population, together with the problems expressed above, can represent a sufficient level of novelty for the advancement of our knowledge in this field.

Reviewer #4: This is a very interesting cross sectional study assessing the prevalence of RLS and sleep disorders in patients with RA exploring various potential contributing factors.

This is a well designed study with vigorous methodology.

The statistical analysis is sophisticated and appropriate.

Major issue:

the paper is written in a form of English that even though are grammatically correct, do not match a scientific way of writing (e.g. we also analyzed, we next compared, etc)

The discussion is very descriptive with no information regarding potential mechanisms and physiological explanations.

Authors need to critically review their data suggesting potential mechanisms of causes

6. PLOS authors have the option to publish the peer review history of their article (what does this mean?). If published, this will include your full peer review and any attached files.

Reviewer #1: No

Reviewer #2: No

Reviewer #3: No

Reviewer #4: No

---

## [Author Response · Author response to Decision Letter 0]

14 Jan 2020

Please see the response to reviewers file.

---

## [Decision Letter · Decision Letter 1]

4 Feb 2020

PONE-D-19-29396R1

The prevalence of insomnia and restless legs syndrome among Japanese outpatients with rheumatic disease: A cross-sectional study

PLOS ONE

Dear Dr. Ichinose,

Thank you for submitting your manuscript to PLOS ONE. After careful consideration, we feel that it has merit but does not fully meet PLOS ONE’s publication criteria as it currently stands. Therefore, we invite you to submit a revised version of the manuscript that addresses the points raised during the review process.

The manuscript improved as a result of the revision. However, the following issues have not been sufficiently addressed:

- 1. Consecutive enrollment should be mentioned in the manucript text and not only in the rebuttal.

- 2. The meaning of the sentence " There was also a lack of generalizability, because all of the rheumatic 

diseases were included for the analyses, and it is difficult to generalize in regard to disease 

backgrounds and treatments associated with insomnia and RLS" is unclear. Generalizability from the STROBE checklist perspective refers to external validity. How does this relate to "generalize in regard to disease 

backgrounds and treatments associated with insomnia and RLS"?

 - 3. The meaning of the sentence "In addition, the total sample size at the start of the study was not determined, because the significance of what our findings revealed related to our outcome measurement (efficacy of the protocol or proposed technique) and the comparisons thereof is exploratory and not confirmative" is even less clear. Please, reword, and make specific reference to statistical power analysis.

 - 4. In the conclusions, please make it explicit that the results mentioned from line 388 onwards are to be considered as exploratory due to the limitations of the study.

We would appreciate receiving your revised manuscript by Mar 20 2020 11:59PM. To enhance the reproducibility of your results, we recommend that if applicable you deposit your laboratory protocols in protocols.io, where a protocol can be assigned its own identifier (DOI) such that it can be cited independently in the future. For instructions see: http://journals.plos.org/plosone/s/submission-guidelines#loc-laboratory-protocols

We look forward to receiving your revised manuscript.

Kind regards,

Alessandro Silvani, M.D., Ph.D.

Academic Editor

PLOS ONE

Reviewers' comments:

Reviewer's Responses to Questions

**Comments to the Author**

1. If the authors have adequately addressed your comments raised in a previous round of review and you feel that this manuscript is now acceptable for publication, you may indicate that here to bypass the “Comments to the Author” section, enter your conflict of interest statement in the “Confidential to Editor” section, and submit your "Accept" recommendation.

Reviewer #1: All comments have been addressed

Reviewer #2: All comments have been addressed

Reviewer #3: (No Response)

2. Is the manuscript technically sound, and do the data support the conclusions?

Reviewer #1: (No Response)

Reviewer #2: Yes

Reviewer #3: No

3. Has the statistical analysis been performed appropriately and rigorously? 

Reviewer #1: (No Response)

Reviewer #2: I Don't Know

Reviewer #3: I Don't Know

4. Have the authors made all data underlying the findings in their manuscript fully available?

Reviewer #1: (No Response)

Reviewer #2: Yes

Reviewer #3: No

5. Is the manuscript presented in an intelligible fashion and written in standard English?

Reviewer #1: (No Response)

Reviewer #2: Yes

Reviewer #3: No

6. Review Comments to the Author

Reviewer #1: (No Response)

Reviewer #2: (No Response)

Reviewer #3: The authors have made little significant changes to their manuscript and my previous concerns have not been addressed adequately.

7. PLOS authors have the option to publish the peer review history of their article (what does this mean?). If published, this will include your full peer review and any attached files.

Reviewer #1: Yes: Monica Puligheddu

Reviewer #2: No

Reviewer #3: No

---

## [Editor Report · Decision Letter 2]

24 Feb 2020

PONE-D-19-29396R2

The prevalence of insomnia and restless legs syndrome among Japanese outpatients with rheumatic disease: A cross-sectional study

PLOS ONE

Dear Dr. Ichinose,

Thank you for submitting your manuscript to PLOS ONE. After careful consideration, we feel that it has merit but does not fully meet PLOS ONE’s publication criteria as it currently stands. Therefore, we invite you to submit a revised version of the manuscript that addresses the points raised during the review process.

In particular, the text at line 69 ("We collected the consecutive data of 121 patients...") is unclear. Does the adjective "consecutive" refer to the data or to the patients? This has implications (cf, eg, paper PMID 23846607). Please, reword and clarify.  

Moreover, the revised text at lines 189-191 and 342-347 appears statistically inconsequent (cf, eg, papers PMID 11560206 and https://doi.org/10.1080/19312450701641375). I suggest that you delete this text and simply modify the text at lines 341-342 as follows: "The limitations of our study deserve some discussion. First, the study population consisted of a small number of outpatients (n=70) treated at a single center, with no a-priori statistical power analysis."

We would appreciate receiving your revised manuscript by Apr 09 2020 11:59PM. To enhance the reproducibility of your results, we recommend that if applicable you deposit your laboratory protocols in protocols.io, where a protocol can be assigned its own identifier (DOI) such that it can be cited independently in the future. For instructions see: http://journals.plos.org/plosone/s/submission-guidelines#loc-laboratory-protocols

We look forward to receiving your revised manuscript.

Kind regards,

Alessandro Silvani, M.D., Ph.D.

Academic Editor

PLOS ONE

---

## [Editor Report · Decision Letter 3]

26 Feb 2020

The prevalence of insomnia and restless legs syndrome among Japanese outpatients with rheumatic disease: A cross-sectional study

PONE-D-19-29396R3

Dear Dr. Ichinose,

We are pleased to inform you that your manuscript has been judged scientifically suitable for publication and will be formally accepted for publication once it complies with all outstanding technical requirements.

With kind regards,

Alessandro Silvani, M.D., Ph.D.

Academic Editor

PLOS ONE
---

## [Editor Report · Acceptance letter]

2 Mar 2020

PONE-D-19-29396R3 

The prevalence of insomnia and restless legs syndrome among Japanese outpatients with rheumatic disease: A cross-sectional study 

Dear Dr. Ichinose:

I am pleased to inform you that your manuscript has been deemed suitable for publication in PLOS ONE. Congratulations! Your manuscript is now with our production department. 

With kind regards,

on behalf of

Prof. Alessandro Silvani 

Academic Editor

PLOS ONE